# The Role of the Unfolded Protein Response on Renal Lipogenesis in C57BL/6 Mice

**DOI:** 10.3390/biom11010073

**Published:** 2021-01-07

**Authors:** Elizabeth Figueroa-Juárez, Lilia G. Noriega, Carlos Pérez-Monter, Gabriela Alemán, Rogelio Hernández-Pando, Ricardo Correa-Rotter, Victoria Ramírez, Armando R. Tovar, Iván Torre-Villalvazo, Claudia Tovar-Palacio

**Affiliations:** 1Metabolic Research Laboratories, Institute of Metabolic Science, Addenbrooke’s Hospital, University of Cambridge, Cambridge CB2 1TN, UK; eif20@medschl.cam.ac.uk; 2Nefrología y Metabolismo Mineral, Instituto Nacional de Ciencias Médicas y Nutrición, Salvador CDMX 14080, Mexico; correarotter@gmail.com; 3Fisiología de la Nutrición, Instituto Nacional de Ciencias Médicas y Nutrición, Salvador CDMX 14080, Mexico; lilia.noriegal@incmnsz.mx (L.G.N.); gabriela.alemane@incmnsz.mx (G.A.); armando.tovarp@incmnsz.mx (A.R.T.); 4Gastroenterología, Instituto Nacional de Ciencias Médicas y Nutrición, Salvador CDMX 14080, Mexico; carlos.perezm@incmnsz.mx; 5Patología Experimental, Instituto Nacional de Ciencias Médicas y Nutrición, Salvador CDMX 14080, Mexico; rogelio.hernandezp@incmnsz.mx; 6Cirugía Experimental, Instituto Nacional de Ciencias Médicas y Nutrición, Salvador CDMX 14080, Mexico; victoria.ramirezg@incmnsz.mx; 7División de Nutrición, Instituto Nacional de Ciencias Médicas y Nutrición, Salvador CDMX 14080, Mexico

**Keywords:** kidney, ER stress, lipogenesis

## Abstract

Renal injury observed in several pathologies has been associated with lipid accumulation in the kidney. While it has been suggested that the accumulation of renal lipids depends on free fatty acids released from adipose tissue, it is not known whether in situ renal lipogenesis due to endoplasmic reticulum (ER) stress contributes to kidney injury. The aim of the present study was to elucidate the role of pharmacological ER stress in renal structure and function and its effect on renal lipid metabolism of C57BL/6 mice. ER stress increased serum creatinine and induced kidney structural abnormalities. Tunicamycin-administered mice developed hyperinsulinemia, augmented lipolysis and increased circulating leptin and adiponectin. Renal unfolded protein response (UPR) gene expression markers, the lipogenic transcription factor SREBP1 and the phosphorylation of eIF2α increased 8 h after tunicamycin administration. At 24 h, an increase in BiP protein content was accompanied by a reduction in p-eIF2α and increased SREBP-1 and FASn protein content, in addition to a significant increase in triglyceride content and a reduction in AMPK. Thus, ER stress induces in situ lipid synthesis, leading to renal lipid accumulation and functional alterations. Future pharmacological and/or dietary strategies must target renal ER stress to prevent kidney damage and the progression of metabolic diseases.

## 1. Introduction

Recent studies have associated abnormal lipid accumulation in the kidney, known as fatty kidney [1], with kidney disease [1,2,3,4,5]. This abnormality has emerged as a common alteration in chronic kidney diseases and acute kidney injury [6,7,8,9]. Renal damage associated with obesity involves uncontrolled free fatty acid (FFA) release from hypertrophic adipose tissue, where its over-accumulation in renal tissue may induce cellular damage [10,11,12,13]. Increasing evidence suggests that renal ectopic lipid accumulation along with altered lipid metabolism triggers kidney damage through lipotoxicity [14,15,16,17]. A study conducted by Koh et al. reported that in diabetic mice, renal AMP-activated protein kinase (AMPK) and its active phosphorylated state are downregulated, leading to a reduction in fatty acid oxidation [18]. Moreover, protein abundance of the lipogenic transcription factor sterol regulatory element-binding protein 1 (SREBP-1) increases in the kidneys of obese mice, exacerbating lipid accumulation [17,18,19]. These alterations have been associated with obesity-related glomerulopathy, hyperfiltration, tubulointerstitial injury and fibrosis in various experimental and human studies [20,21].

Endoplasmic reticulum (ER) stress is an important underlying mechanism in several renal alterations that may induce glomerular and tubular damage, as this has been recently observed in humans and in animal models [22,23,24,25]. ER stress is a cellular mechanism activated in response to diverse insults, such as nutrient and oxygen deprivation, unfolded and misfolded proteins in the ER, oxidative stress and nutrient excess [22,26,27]. To restore cellular homeostasis, the ER activates a complex signaling pathway aimed to reestablish normal cellular functions called the unfolded protein response (UPR) [28,29,30].

In mammalian cells, the UPR is composed of three major branches, including PKR-like ER kinase (PERK), inositol-requiring enzyme 1 (IRE-1) and activating transcription factor 6 (ATF6) [28,31]. One purpose of the UPR is to shut off general protein translation, reducing the flux of proteins into the ER. The PERK pathway phosphorylates and inactivates elongation initiation factor 2 alpha (eIF2alpha), halting protein synthesis [26,32], the ATF6 pathways upregulate the expression of ER chaperones such as BiP, promoting proper protein folding, and the IRE-1 alpha pathway activates ER-associated degradation (ERAD) and X-box binding protein 1 (XBP1) splicing and translation in order to reestablish cellular homeostasis [33]. However, when ER stress is not resolved, the UPR can induce cell death [34]. In the liver, ER stress increases lipogenesis by the upregulation of genes involved in lipid metabolism, inducing hepatic steatosis [35]. However, it is not known whether ER stress in the kidney also induces in situ lipid accumulation and alters kidney function. In order to evaluate the direct effect of ER stress in renal lipid metabolism, mice without any other physiological alterations such as obesity or diabetes were administered the N-linked glycosylation inhibitor tunicamycin, which induces ER stress, and euthanized at 8 or 24 h after ER stress induction in order to evaluate time-dependent ultrastructural morphology changes in renal tissue, renal lipid metabolism and AMPK activity.

## 2. Materials and Methods

### 2.1. Mice and Experimental Protocol

Male C57BL/6 mice (*n* = 6) aged 6–7 weeks old were obtained from the experimental research department of our institute and were maintained in micro-isolator cages on a 12-h light/dark cycle. All mice were given free access to food and water. To induce pharmacological ER stress, mice were administered a single intraperitoneal injection of 100 μg/mL tunicamycin (Sigma, St. Louis, MO, USA) diluted in 150 mM dextrose at 1 g/Kg body weight [36]. The control group (C) was injected with vehicle. Experimental groups were administered with tunicamycin and euthanized 8 h (T8) or 24 h (T24) after injection (*n* = 6). Tunicamycin is a competitive inhibitor of the N-acetylglucosamine phosphotransferase, blocking N-glycosylation in the ER. Impaired N-glycosylation prevents adequate protein folding in the ER, leading to misfolded protein accumulation and activation of the UPR. The timing of sample collection was based on previous work indicating that at 8 h, UPR activation is mostly at the mRNA level, whereas at 24 h, the increase is at the protein level [37,38,39]. At the end of the study, mice were food-deprived for 6 h and euthanized by CO_2_ asphyxiation. All animals were allocated into metabolic cages 8 h before euthanasia for urine collection. Body weights were measured just before CO_2_ asphyxiation. The right kidney was removed and weighed, quickly frozen in liquid nitrogen and stored at −70 °C for molecular studies. Blood was obtained from the posterior vena cava using a 1 mL syringe. Serum was separated by centrifugation for 10 min at 1800× *g* at 4 °C. One half of the left kidney was blanched with phosphate buffer and fixed in 10% formalin. The other half was fragmented into small pieces, immersed in 10% glutaraldehyde in cacodylate buffer pH 7.2 until fixation was completed for histological analysis. All animal procedures were conducted in conformity with the Guide for the Care and Use of Laboratory Animals of the NIH [40]. The protocol was submitted and approved by the institutional animal ethics research committee (FNU-1917-18-20-1).

### 2.2. Renal Function Parameters

Serum creatinine, blood urea nitrogen (BUN) and serum uric acid were measured with colorimetric kits in an automatized analyzer (UniCel DxC 600, Beckman Coulter, Indianapolis, IN, USA). Urinary H_2_O_2_ excretion was determined by a fluorescent method using Amplex Red (Invitrogen) [41]. The fluorescence was measured using excitation and emission wavelengths of 571 and 585 nm, respectively, using a Synergy HT multimode microplate reader (Biotek Instruments, Winooski, VT, USA).

### 2.3. Histological Analysis

The left kidney of each mouse was rapidly removed, sliced longitudinally and immersed in ice-cold 10% paraformaldehyde in phosphate-buffered saline. After dehydration, kidney slices were embedded in paraffin, sectioned at 4 µm and stained using hematoxylin and eosin. Glomerular size and urinary space were measured in tissue sections stained with H&E. The outline of 20 glomeruli from each animal was digitalized from the light microscope using a video camera and a computer-based analysis system (magnification 200%) (Q/win-500, Leica, Milton Keynes, UK). Glomerular sections were displayed on the computer screen, and their area was measured by an interactive procedure with an image analysis software package. To normalize the structure for measuring, only glomeruli in which both poles, vascular and tubular, were clearly seen were taken into account to perform these determinations. To visualize neutral lipids, frozen kidney sections (8 µm) were stained with Oil Red O (ORO; Sigma, St. Louis, MO, USA) and counterstained with hematoxylin.

### 2.4. Apoptosis Analysis

Apoptosis was determined by using a terminal deoxynucleotidyl transferase-mediated (TUNEL) assay. Detection of apoptotic cells was performed in 5 µm of formalin-fixed, paraffin-embedded renal tissue by in situ TUNEL ApopTag^®^ Plus Peroxidase, In Situ Apoptosis Detection Kit (Chemicon-Millipore, Billerica, MA, USA) according to the manufacturer’s instructions. Cells were quantified in 6 fields of each mouse (magnification 40×). TUNEL-positive cells were considered those that had nuclear condensation brown staining and quantified as positive cells per square pixel. Results were presented as an average of apoptotic cells ± SE per group with respect to the control mice, considering the control group as 100%.

### 2.5. Electron Microscopy

To study the ultrastructural morphology of convoluted proximal renal tubules, small kidney tissue fragments from the cortex were immersed in 10% glutaraldehyde dissolved in cacodylate buffer pH 7.2 and post-fixed with osmium tetroxide, dehydrated in graded ethyl alcohol solutions and embedded in Epon resin (London Resin Company, London, UK). Thin sections from 70 to 90 nm were placed on copper grids, contrasted with lead and uranium salts and examined with an FEI Technei electron microscope.

### 2.6. Biochemical Determinations

Serum glucose was assayed using enzymatic kits (DiaSys Diagnostic Systems GmbH, Holzheim, Germany). The serum free fatty acid concentration was assayed with a Roche FFA kit (Roche Applied Science). Serum glycerol was measured using the Free Glycerol Reagent (Sigma). Serum insulin, leptin and adiponectin concentrations were measured by radioimmunoassay (Millipore).

### 2.7. RNA Extraction and mRNA Quantitation

Total RNA from kidney was extracted with TRIzol reagent (Invitrogen, Carlsbad, CA, USA) following the manufacturer’s protocol. The RNA concentration was determined spectrophotometrically with a NanoDrop (Thermo Scientific, Wilmington, DE, USA). cDNA synthesis was performed with 3 µg of total RNA using Oligo (dt)_12–18_ and M-MLV reverse transcriptase (Invitrogen), according to the manufacturer’s recommendations. All genes were quantified using TaqMan assays (Applied Biosystems, Foster City, CA, USA). TaqMan-based real-time quantification was performed in a LightCycler 480 Instrument (Roche Molecular Biochemicals, Indianapolis, IN, USA). The cycling conditions were based on the instructions provided by the manufacturer. The reference assay ID for each gene is as follows: Kim-1 (Mm00506686_m1), ATF-6 (Mm01295317_m1), BiP (Mm00517691_m1), CHOP (Mm00492097_m1), SREBP-1 (Mm00550338_m1), FAS (Mm00662319_m1) and Cyclophilin (Mm02342429_g1). For all genes, samples were run in triplicate in 96-well reaction plates. Data were analyzed using LightCycler 480 software, and Ct was calculated by the second derivate maximum method. Target gene expression levels were normalized to cyclophilin mRNA abundance as the internal invariant control.

### 2.8. Protein Extraction, SDS/PAGE and Immunoblotting

Frozen tissues were homogenized in ice-cold RIPA buffer with a protease inhibitor cocktail (Complete, Roche Applied Science) using a Kontes Pellet Pestle mixer (Thermo Fisher Scientific, Rockford, IL, USA). The protein concentration was determined using the DC protein assay kit (Bio-Rad Laboratories, Richmond, CA, USA). Tissue lysates (40 µg) were combined with Laemmli sample buffer and separated by SDS-PAGE. After electrophoretic separation, the proteins were electrotransferred to a PVDF membrane using a Mini Trans-Blot Electrophoretic Transfer Cell (Bio-Rad), blocked and incubated overnight at 4 °C with the following antibodies: ß-Actin, CHOP, ATF6, BiP, FASn, AMPK, p-AMPK (thr 172) (Santa Cruz Biotechnology, Santa Cruz, CA, USA), eIF2α and p-eIF2α (ser 51) (Millipore-Upstate, Billerica, MA, USA), IRE-1 and SREBP-1 (Abcam, Cambridge, MA, USA). After incubation with secondary antibodies conjugated to horseradish peroxidase, chemiluminescence detection was carried out using a chemiluminescent Western blotting kit (Immobilon Western Chemiluminescent HRP Substrate, Millipore). Digital images of the membranes were obtained by a ChemiDoc MP densitometer and processed by Image Lab software (Bio-Rad). The results are reported relative to ß-actin. A value of 1 was arbitrarily assigned to the control samples, which were used as a reference for the other conditions.

### 2.9. Kidney Lipid Content

Total lipids were extracted from kidney tissue according to Folch et al. [42]. The organic layer was dried under nitrogen gas and solubilized in isopropanol/Triton X-100 (10%) and then assayed for triglyceride concentration using an enzymatic kit (DiaSys). Data were expressed as the amount of triglycerides per gram of wet weight.

### 2.10. Statistical Analysis

Data are presented as the mean ± SEM. Statistical analysis was performed using one-way ANOVA with a post hoc test using Fisher’s LSD method for multiple comparisons, using Prism 8 for Mac (GraphPad, San Diego, CA, USA). Differences were considered statistically significant at *p* < 0.05 and are indicated by letters in the figures (a > b > c).

## 3. Results

### 3.1. ER Stress Is Associated with Alterations in Kidney Function and Structure

To understand whether the presence of ER stress is associated with renal dysfunction, we induced ER stress in mice through a single i.p. low dose of tunicamycin and observed the metabolic and morphological alterations after 8 and 24 h. We observed an increase in serum creatinine, after 8 and 24 h of tunicamycin administration. However, serum uric acid decreased after 8 h and increased significantly after 24 h compared to 8 h. BUN showed a slight increase without statistical significance (Figure 1A–C). The kidney injury molecule 1 (Kim-1), an acute injury marker in renal proximal tubules, was analyzed in order to determine if the induced ER stress modified this biomarker. Results from the Kim-1 PCR showed that the low dose of tunicamycin administration significantly increased Kim-1 mRNA expression in the 24 h group, confirming that ER stress induced by tunicamycin caused tubular injury (Figure 1D). Additionally, the urinary concentration of hydrogen peroxide was evaluated in order to determine if a reactive molecule derived from molecular oxygen as part of ROS molecules was present. After 8 h of tunicamycin treatment, a significant increase in oxidative stress was observed that was partially reduced at 24 h (Figure 1E). Histopathological examination by H&E staining showed that an acute exposition to tunicamycin modifies the kidney tubular structure compared with vehicle-treated mice. We observed that ER induced a significant reduction in the diameter of glomeruli as well as a decrease in the urinary space, suggesting that tunicamycin not only induced tubular damage but also modified the glomerular structure (Figure 1F,G). Eight hours after tunicamycin administration, few tubular lesions were observed, but at 24 h, lesions increased. The lesions observed were loss of the epithelial brush border and accumulation of debris in the luminal area, as well as vacuole formation (Figure 1H). To evaluate renal lipid infiltration, frozen kidney sections were stained with Oil Red O. As observed in Figure 2A–C, control mice did not show intracellular lipids nor visible ultrastructural alterations. Interestingly, renal ER stress induced lipid droplet accumulation, particularly in the renal cortex (Figure 2D). Electron microscopy study confirmed the existence of cytoplasmic lipid vacuoles, dilatation of the rough endoplasmic reticulum and an increase in lysosomes 8 h after tunicamycin administration (Figure 2E,F). After 24 h, numerous epithelial cells in the proximal convoluted tubules presented lipid droplets (Figure 2G). Low- and high-power electron microscopy micrographs revealed an increased number of dead cells specifically in the proximal convoluted tubule (Figure 2H,I). This finding was confirmed by the TUNEL assay, where we observed that tunicamycin induced apoptosis after 8h of administration, but this effect was exacerbated after 24 h (Figure 3A,B). In the TUNEL assay, positive cells are shown by intense brown nuclear staining. On the other hand, other cells showed cytoplasmic dissolution at the base in addition to mitochondria with different abnormalities, such as matrix expansion, broken cristae and electron-dense deposits. Some lysosomes were associated with empty vacuoles, forming incipient autophagosomes (Figure 2H,I). Thus, endoplasmic reticulum stress in the kidney was accompanied by an acute reduction in renal function and an altered morphology. Most importantly, the presence of lipid droplets was observed as a distinctive characteristic that ought to be studied further.

### 3.2. ER Stress Alters the Metabolic and Endocrine Parameters of Mice Treated with Tunicamycin

Endoplasmic reticulum stress has been identified as the underlying mechanism leading to the development of metabolic alterations in several organs [43]. Thus, we evaluated the effect of the acute induction of ER stress in systemic metabolism. We found no significant changes in either body or kidney weight after tunicamycin administration, maintaining the kidney weight/body weight ratio (Table 1). However, tunicamycin-induced ER stress significantly reduced serum glucose, particularly after 24 h.

Interestingly, serum insulin dramatically increased by 1.9- and 3.2-fold after 8 and 24 h, respectively. In addition, FFA and glycerol also significantly increased by 1.1- and 0.7-fold 8 h after tunicamycin administration and 0.6- and 1.0-fold after 24 h. Circulating leptin and adiponectin concentrations increased by 4.5- and 0.7-fold, respectively, 8 h after tunicamycin administration. After 24 h, the circulating content of these adipokines diminished even though their levels remained significantly higher than in control mice (Table 1). These results suggest that acute ER stress induced by tunicamycin causes hyperinsulinemia, increased lipolysis and adipokine secretion from adipose tissue.

### 3.3. Tunicamycin Causes ER Stress in the Kidney

To evaluate the direct effect of a low-dose tunicamycin injection in the induction of ER stress markers in the kidney, we evaluated the expression of several UPR markers at 8 and 24 h after tunicamycin administration. In response to increased protein load in the ER, the transcription factor XBP-1 is activated by mRNA splicing by IRE-1. XBP-1 along with the chaperone BiP is involved in the restoration of cellular homeostasis in response to acute ER stress. Our data clearly show that 8 h after tunicamycin administration, spliced XBP-1 and BiP mRNA rapidly increased by 12.4- and 31.6-fold, respectively (*p* < 0.001) (Figure 4A,B). Protein abundance of BiP presented a time-dependent increase by 14.5- and 88-fold after 8 h and 24 h, respectively (Figure 4C,D). The time-dependent increase in BiP protein content despite a reduction in its mRNA abundance at 24 h reflects the selective translation of high-priority proteins involved in the restoration of cellular homeostasis. Accordingly, protein abundance of IRE1 also increased in a time-dependent manner, indicating increased XBP-1 mRNA processing (Figure 4D,E). The mRNA and protein abundance of ATF6 and CHOP increased at 8 h as a response to pharmacologic ER stress but both mRNA and protein abundance were reduced at 24 h after tunicamycin exposure (Figure 4D,F–I). Consistently, phosphorylation of the translation initiation factor eIF2α increased 8 h after tunicamycin administration, indicative of a blockade of global protein synthesis. As observed with ATF6 and CHOP, eIF2α phosphorylation was reduced at 24 h to a level similar to that in control mice (Figure 4D,J). These data clearly demonstrate that in response to impaired ER folding capacity due to tunicamycin administration, the kidney develops severe ER stress at 8 h which is partially resolved after 24 h due to activation of the adaptive UPR.

### 3.4. ER Stress Associates with Renal Lipogenesis

We evaluated the effect of the acute induction of ER stress in lipid metabolism in the kidney. Kidney ER stress was accompanied by increased expression of genes involved in lipid synthesis, indicating that the kidney presents exacerbated lipogenesis in this condition. In fact, 8 h after tunicamycin administration, the mRNA content of the transcription factor SREBP-1 was significantly higher than in vehicle-treated mice, as was the mRNA content of FASn, one of its target genes (Figure 5A,B). This upregulation was associated with a significant increase in renal triglyceride content (Figure 5C). In fact, Oil Red O staining revealed ectopic lipid accumulation, particularly in the proximal convoluted renal tubules (Figure 2A,D). Western blot analysis revealed a similar trend in SREBP-1 protein abundance. In contrast, the FASn protein content was significantly reduced 8 h after tunicamycin administration but increased to a level higher than in control mice at 24 h (Figure 5D,E,G). These results indicated that SREBP-1 mRNA transcription and translation were favored during ER stress, although FASn translation was probably halted at 8 h in response to eIF2α phosphorylation. Interestingly, both total and phosphorylated AMPK were significantly reduced in response to tunicamycin administration in a time-dependent fashion (Figure 5F,G). AMPK is an intracellular energy sensor that induces mitochondrial biogenesis and fatty acid oxidation in response to energy stress. Thus, the reduction in AMPK abundance and activity during ER stress indicates an impairment in fatty acid oxidation and energy metabolism in kidneys subjected to pharmacological ER stress, favoring triglyceride accumulation and lipotoxicity.

## 4. Discussion

Accumulating evidence indicates that ER stress contributes to glomerular and tubular damage [22,44] and increases lipid synthesis in several organs [45]. We report here that pharmacologically induced ER stress activates the UPR in the kidney in a time-dependent manner. The expression of UPR and lipogenic genes increased significantly during the first 8 h after tunicamycin administration. As expected, there was an increase in the phosphorylation state of eIF2α, halting protein synthesis, and thus lipogenic protein abundance did not show a parallel increase with the corresponding mRNAs at that time. These changes were accompanied by maintenance of the phosphorylated active form of AMPK, possibly increasing the use of triglycerides as energy fuel to cope with the increased energy demand posed by ER stress, avoiding lipid accumulation. After 24 h, the amount of the chaperone BiP and transcription factor XBP1-1 greatly increased to restore ER homeostasis, accompanied by a reduction in the phosphorylated form of eIF2α, allowing the translation of most proteins. These changes were accompanied with a significant increase in SREBP-1 and FASn protein abundance and a reduction in AMPK phosphorylation, indicating a shift in renal lipid metabolism from oxidation to synthesis, leading to increased accumulation of triglycerides in the kidney.

The present study demonstrates that pharmacologically induced ER stress is associated with acute alterations in kidney function and structure. Serum creatinine concentrations significantly increased in the tunicamycin-injected group after 24 h. Higher serum creatinine is in agreement with previous studies, where tunicamycin increases serum creatinine levels, in particular after 24 h of the administration of tunicamycin [46,47]. Interestingly, Liu et al. [48] found in female mice that tunicamycin at a lower dose not alter kidney function when the mice are young, but serum creatinine and BUN levels are significantly increased when the mice age. Therefore, the gender and age of mice induced different responses to tunicamycin treatment. On the other hand, uric acid showed a decrease after 8 h of tunicamycin treatment compared with levels founded after 24 h. Elevated serum uric acid is a strong predictor of the development of fatty liver as well as metabolic syndrome. A study conducted by Choi et al. in 2014 [49] demonstrated that uric acid induced ER stress in hepatocytes, leading to SREBP-1c activation and triglyceride accumulation. These events could occur in the kidney, since increasing evidence suggests that hyperuricemia is an independent risk factor for the occurrence and development of kidney disease. In a study conducted by Li et al., 2013, the authors identified that uric acid induced changes in rat glomerular mesangial cells via ER stress [50]. In human renal tubular epithelial cells (HK-2) treated with uric acid, morphological changes were observed and increased inflammatory cytokines were present [51].

Oxidative protein folding in the endoplasmic reticulum is a significant source of hydrogen peroxide (H_2_O_2_), and for correct protein folding, the redox state of the ER must be efficiently regulated. As such, several mechanisms with varying degrees of overlap manage the redox state of the ER [52]. In order to analyze reactive molecules derived from molecular oxygen, we evaluated hydrogen peroxide as a marker of ROS production. Interestingly, hydrogen peroxide was significantly increased after 8 h of tunicamycin treatment, and after 24 h, it returned to levels observed in the control group. This finding indicates that hydrogen peroxide probably increased due to an uncontrolled ROS load by tunicamycin that would lead to ER stress activating the unfolded protein response. In order to prevent ROS overload, the ER needs a robust antioxidant mechanism.

A high-sensitivity and specific biomarker of acute kidney injury in early diagnosis is Kim-1, a transmembrane glycoprotein [53], which is highly expressed in the differentiation, proliferation and early damage of proximal renal tubular epithelial cells. In our study, Kim-1 was significantly upregulated after 24 h of tunicamycin treatment. Even though no statistical significance was observed between the control and 8 h tunicamycin treatment groups, Kim-1 gene expression was three times higher in the 8 h treatment group, indicating an alteration in renal tubular epithelial cells, since no expression should be observed in healthy kidney tissue.

Clear evidence for acute renal damage induced by an acute low dose of tunicamycin is the structural alterations at the cellular level. Histopathological evaluation of renal microscopic images indicates that the glomerulus diameter and Bowman’s space undergo alterations after tunicamycin exposure. Glomeruli diameter was reduced after 8 and 24 h post-tunicamycin treatment, suggesting a possible role of the UPR in deactivation and redifferentiation of mesangial cells. Previous reports have suggested that the UPR has the potential to inhibit cell proliferation. For example, Brewer and Diehl [54] showed that ER stress caused induction of the PERK-eIF2α pathway and consequent translational suppression of cyclin D1, culminating in G0/G1 cell cycle arrest in NIH3T3 cells [55].

It is worth noting that all these metabolic alterations occurred very rapidly in response to acute ER stress. Indeed, no significant differences in the composition of body weight or kidney weight were noted among groups during the 8 or 24 h tunicamycin challenge. Despite the absence of changes in body weight, acute ER stress could alter the metabolic activities of several organs, such as adipose tissue. As reported previously, acute ER stress impairs adipose tissue endocrine functions [56]. The adipokines adiponectin and leptin exert important actions in the kidney and thus adipokine dysregulation during ER stress may potentiate ER stress-induced alterations in kidney function. Adiponectin is cleared rapidly from the circulation primarily by the liver and secondarily by the kidneys [57]. High circulating adiponectin is strongly associated with reduced cardiovascular risk; however, in obesity and type 2 diabetes, hypoadiponectinemia resulting from low adipose tissue adiponectin secretion favors the development of insulin resistance and cardiovascular disease [58]. Chronic kidney disease (CKD) is a unique condition that occurs with exceedingly high insulin resistance and cardiovascular morbidity and mortality [59,60] and is paradoxically associated with elevated plasma adiponectin [58]. In this study, adiponectin levels increased in a time-dependent manner, as a result of increased adiponectin secretion from adipose tissue and probably an ER stress-dependent reduction in kidney capacity for clearing of circulating adiponectin. On the other hand, one of the main actions of leptin in the brain is to regulate food intake [61]. Leptin also stimulates ß-oxidation of fatty acids, as well as export of lipids from tissues through signaling events involving both central (brain) and peripheral receptors [62,63]. Interestingly, ER stress is sufficient to inhibit leptin signaling in cultured cells, whereas pharmacological inhibition of ER stress improves leptin signaling [64,65]. In this study, tunicamycin significantly increased leptin serum levels after 8 h of tunicamycin treatment and remained significantly higher than the control group after 24 h. However, the beneficial or negative effect of increased adiponectin and leptin levels in renal metabolism during acute ER stress needs to be further investigated. It has been reported that ER stress regulates hepatic gluconeogenesis [66,67] and acute tunicamycin treatment reduces blood glucose levels. Accordingly, to these previous observations, we found that 24 h tunicamycin treatment decreased blood glucose (Table 1). The kidney is a gluconeogenic organ and provides approximately 30% of circulating glucose during fasting [68]. The role of renal ER stress in the reduction in circulating glucose during acute tunicamycin challenge is unknown and warrants future studies.

We have previously demonstrated that acute tunicamycin administration induces ER stress in adipocytes, leading to increased lipolysis [56]. Increased circulating FFA could favor fatty infiltration in the kidney. Our results show that intraperitoneal administration of a low dose of tunicamycin rapidly induced an ER stress response with increased circulating FFA and glycerol after 8 and 24 h of tunicamycin treatment. These results are consistent with the previous study by Bogdanovic et al., where they reported that tunicamycin induced an ER stress response in adipose tissue that correlates with increasing circulating FFAs and glycerol. However, Feng et al. [69] and Wu et al. reported a reduction in serum FFA after tunicamycin treatment [36,69]. One possible reason for these discrepancies is the duration of tunicamycin treatment, or the fed or fasted state of mice just before euthanasia [69]. The increased circulating FFA content observed in mice administered with tunicamycin clearly correlates with lipid droplets accumulation in proximal tubular cells and the confirmation of cytoplasmic lipid vacuoles, in addition to the formation of incipient autophagosomes. Mammalian cells activate autophagy to eliminate damaged proteins and organelles by lysosomal degradation. The resulting amino acids, free sugars and fatty acids are recycled to cope with the bioenergetic needs of the cell and for serving as building blocks for new protein and organelle synthesis. Therefore, the process of autophagy provides a protective role against accumulation of defective proteins and organelles and thus the normal flux of the autophagic activity prevents the activation of cell death pathways. However, under certain situations, excessive autophagy induces an impairment in the completion of the autophagy process and over-digestion of cytoplasmic organelles contents, leading to autophagic cell death [70]. Activation of ER stress has been associated with disproportionate autophagy and the development of atherosclerosis and cardiovascular disease [71]. In the present study, renal ER stress induced apoptosis in a time-dependent manner, where epithelial tubular cells were mainly affected. These results suggest that epithelial tubular cells are particularly susceptible to acute ER stress.

Accumulating evidence supports the hypothesis that prolonged metabolic imbalance of lipids leads to ectopic fat distribution in the peripheral organs (lipotoxicity), including the kidney. Despite the significant impact of renal lipid metabolism in the progression of kidney diseases, the pathophysiological consequences of renal lipid accumulation in response to ER stress have only been partially been elucidated. This work paves the way for novel pharmacological and non-pharmacological approaches for the treatment of kidney diseases through mitigation of ER stress and prevention of lipid accumulation (Figure 6).

## 5. Conclusions

Renal oxidative stress, apoptosis and lipotoxicity have been identified as the three main alterations associated with mesangial expansion, tubular damage and glomerulosclerosis. The elucidation of the molecular mechanisms linking renal metabolic alterations with disease progression is fundamental to propose novel clinical biomarkers of early renal damage and to develop innovative therapeutic approaches to prevent progression to advanced stages of renal disease. The results of the present study point to renal endoplasmic reticulum stress as a highly significant mechanism that triggers the three main alterations that lead to kidney disfunction and injury, i.e., (1) mitochondrial dysfunction and reduced fatty acid oxidation, (2) unrestrained autophagia and (3) augmented SREBP-1 activity, increasing fatty acid synthesis. The concurrence of these abnormalities in the kidney induces oxidative stress, apoptosis and lipotoxicity and may lead to chronic kidney alterations (Figure 6). The efficient resolution of ER stress should be an important target for future therapeutic approaches for preventing kidney diseases.

## Figures and Tables

**Figure 1 biomolecules-11-00073-f001:**
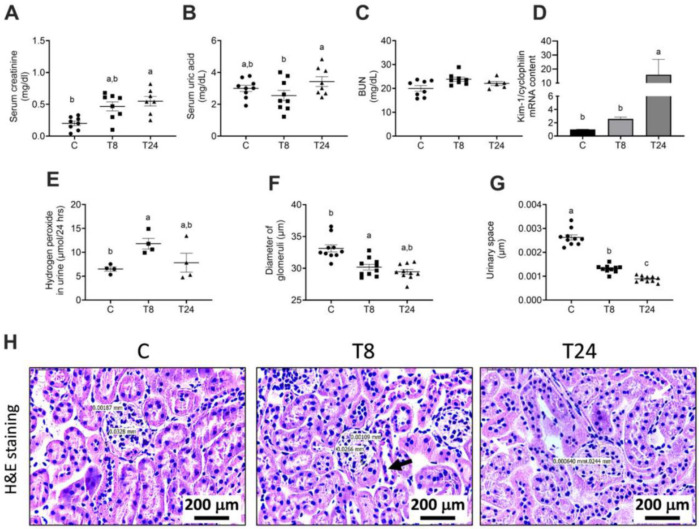
Renal functional alterations associated with endoplasmic reticulum (ER) stress. Mice were administered a single i.p. injection of vehicle or tunicamycin (1 g/Kg body weight), and kidneys were obtained after 8 h (T8) or 24 h (T24). The following graphs display: serum creatinine (**A**), serum uric acid (**B**), BUN (**C**), Kim-1 gene expression (**D**), urinary hydrogen peroxide (**E**), morphometric analysis of glomerular diameter (**F**) and urinary space (**G**) in kidney sections stained with H&E (20× magnification). Values are means ± SEM, *n* = 4–6. Means without a common letter differ significantly, a > b > c (*p* < 0.05). (**H**) Representative micrographs of kidney H/E-stained sections from the different groups, in comparison with the control group (**c**), glomeruli are smaller in T8 and T24 groups, kidney from T8 group shows some proximal cortical tubules revisited by the flat damaged epithelium without apical microvilli (arrow), while kidney from the T24 group exhibits numerous tubular epithelial cells with lipid cytoplasmic vacuoles (arrows).

**Figure 2 biomolecules-11-00073-f002:**
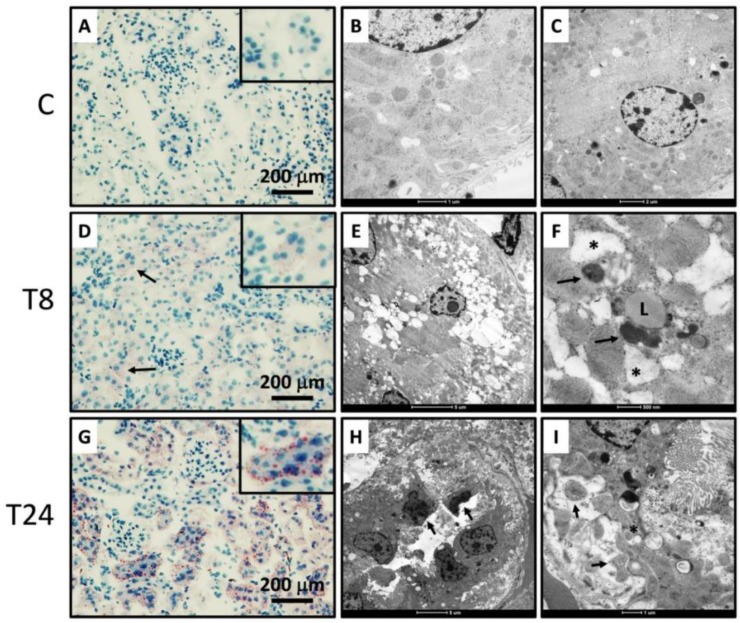
Altered kidney morphology induced by tunicamycin administration. (**A**) Frozen kidney sections of control mice stained with Oil Red O. (**B**) Low-power electron microscopy of kidneys from control mice. (**C**) High-power electron microscopy of kidneys from control mice. (**D**) Frozen kidney sections stained with Oil Red O 8 h after tunicamycin administration. Some convoluted tubules show red stained droplets in the cytoplasm (arrows) that correspond to lipid vacuoles. Inset shows high-power field tubule with some lipid droplets. (**E**) Low-power electron microscopy shows numerous cytoplasmic vacuoles in epithelial cells from convoluted tubules 8 h after tunicamycin administration. (**F**) High-power electron micrograph of proximal convoluted epithelial cell 8 h after tunicamycin administration. The distended rough endoplasmic reticulum (asterisks) is surrounded by mitochondria, lysosomes and lipid droplets (L). (**G**) Frozen section stained with Oil Red 24 h after tunicamycin administration. There are numerous epithelial cells from proximal convoluted tubules with red labeled lipid droplets. Inset shows high-power field tubule with numerous lipid vacuoles. (**H**) Low-power electron microscopy micrograph from proximal convoluted tubule 24 h after tunicamycin administration. There are necrotic cells (arrow) intercalated with damaged epithelial cells with basal cytoplasmic dissolution (asterisks). (**I**) High-power micrograph from the same area shown in H. There is cytoplasmic dissolution with some mitochondria with a slightly expanded matrix (arrows) and lysosomes associated with vesicles conforming to autophagosomes (asterisk).

**Figure 3 biomolecules-11-00073-f003:**
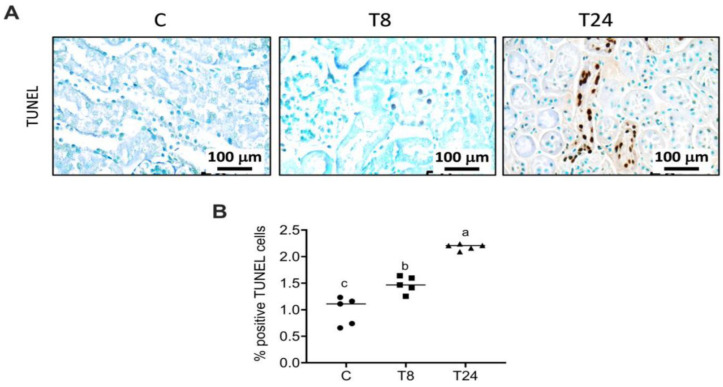
ER stress-induced apoptosis in mouse kidneys. Mice were treated with a single i.p. injection of vehicle or tunicamycin (1 g/Kg body weight), and the kidney was obtained after 8 h (T8) or 24 h (T24). Kidney sections from tunicamycin-treated groups were TUNEL-stained to identify apoptotic cells. Apoptotic cells were not identified in the control group, while apoptotic cells were observed after 8 h and 24 h of tunicamycin treatment (**A**). Assessment of the average percentage of TUNEL-positive cells (**B**). Values are means ± SEM, *n* = 4–6. Means without a common letter differ significantly, a > b > c (*p* < 0.05).

**Figure 4 biomolecules-11-00073-f004:**
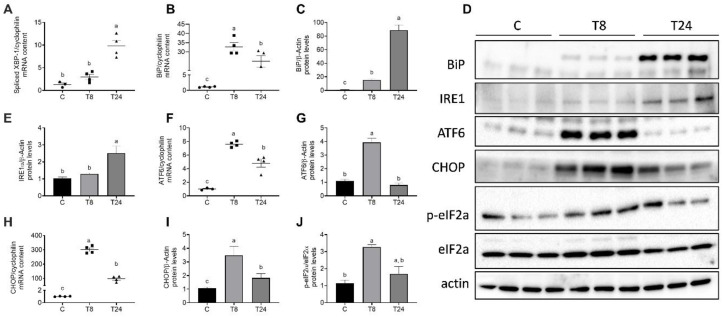
Activation of the unfolded protein response (UPR) in the kidney of mice treated with tunicamycin. Mice received a single i.p. injection of vehicle or tunicamycin (1 g/Kg body weight), and the kidney was obtained after 8 h (T8) or 24 h (T24). Spliced XBP-1 (**A**) and BiP mRNA content (**B**), densitometric analysis of BiP protein content (**C**), BiP, IRE-1, ATF6, CHOP and total and phosphorylated eIF2α immunoblots (**D**), densitometric analysis of IRE-1 protein abundance (**E**), ATF6 mRNA (**F**) and protein abundance (**G**), CHOP mRNA (**H**) and protein abundance (**I**) and densitometric analysis of p-eIF2α/eIF2α, (**J**) Values are means ± SEM, *n* = 4–6. Means without a common letter differ significantly, a > b > c (*p* < 0.05).

**Figure 5 biomolecules-11-00073-f005:**
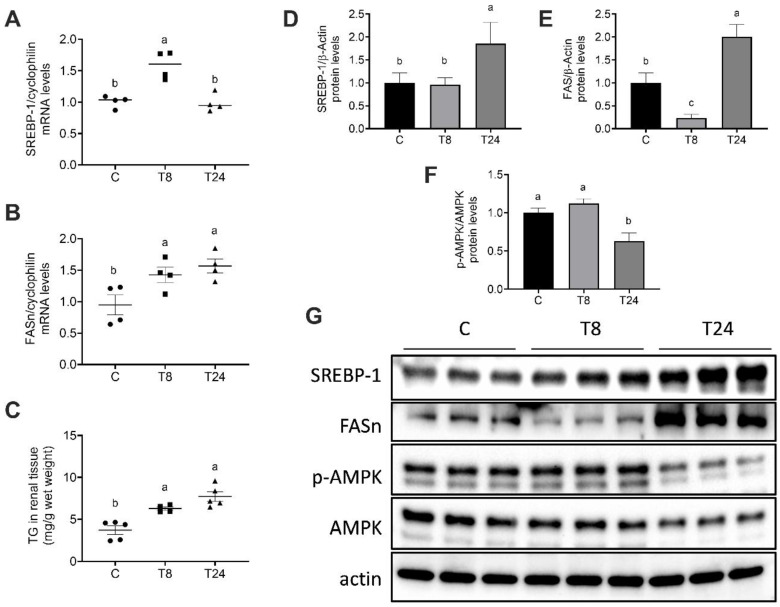
ER stress-induced renal lipogenesis in mice treated with tunicamycin. Mice were treated with a single i.p. injection of vehicle or tunicamycin (1 g/Kg body weight), and kidneys were obtained after 8 h (T8) or 24 h (T24). mRNA content of SREBP-1 (**A**) and FASn (**B**), triglyceride content in renal tissue (**C**), densitometric analysis of the protein content of SREBP-1 (**D**), FASn (**E**) and p-AMPK/AMPK (**F**), protein content of SREBP-1, FASn, AMPK and p-AMPK (thr 172) (**G**). Values are means ± SEM, *n* = 4–6. Means without a common letter differ significantly, a > b > c (*p* < 0.05).

**Figure 6 biomolecules-11-00073-f006:**
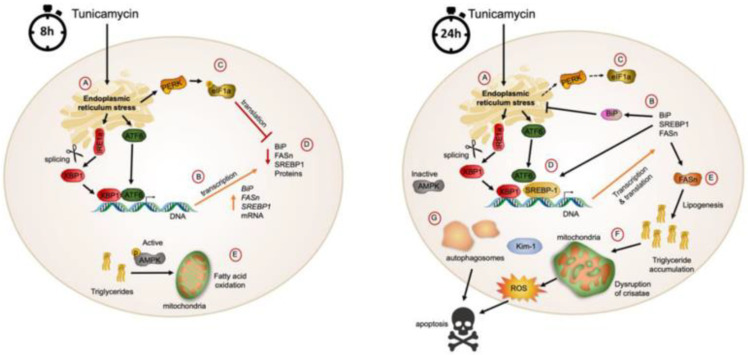
Altered metabolic pathways in response to ER stress in the kidney. Diagram representing the time-dependent role of endoplasmic reticulum stress in renal metabolic alterations. At 8 h (left) after administration, tunicamycin induces ER stress (**A**), increasing sliced XBP-1, BiP, SREBP-1 and FASn transcription (**B**). However, the inactivation of eIF2α reduc-es general protein translation (**C**,**D**). At this time point, AMPK is still active and stimulates triglyceride hydrolysis and mitochondrial fatty acid oxidation (**E**). At 24 h (right), ER stress (**A**) augments BiP transcription and translation (**B**), re-ducing PERK-induced eIF2α phosphorylation (**C**), increasing SREBP1 transcriptional activity (**D**) and FASn translation (**E**) along with a reduction in AMPK activity. As a result, renal lipid accumulation increases, leading to lipotoxicity and an impairment in the mitochondrial structure (**F**). Dysfunctional mitochondria increase the release of radical oxygen species, inducing cellular damage and activating Kim-1 which induces autophagy (**G**). However, excessive autophagy leads to apoptosis and a reduction in renal function.

**Table 1 biomolecules-11-00073-t001:** Body weight, kidney weight and biochemical parameters in mice treated with tunicamycin for 8 h and 24 h.

Parameter	Control	Tunicamycin 8 h	Tunicamycin 24 h
Body weight, (g)	29.0 ± 1.6	32.3 ± 1.4	29.0 ± 0.8
Kidney weight, (g)	0.196 ± 0.02	0.230 ± 0.01	0.205 ± 0.01
Kidney weight/body weight (%)	0.67	0.71	0.72
Glucose, mg/dl	232 ± 9.7 ^a^	205 ± 15.0 ^a,b^	170.8 ± 13.4 ^b^
Insulin (ng/mL)	2.01 ± 0.12 ^c^	6.00 ± 0.22 ^b^	8.44 ± 0.51 ^a^
NEFA (mM)	1.81 ± 0.1 ^c^	3.79 ± 0.1 ^a^	2.89 ± 0.3 ^b^
Glycerol (mM)	0.15 ± 0.004 ^b^	0.26 ± 0.001 ^a^	0.30 ± 0.002 ^a^
Leptin (ng/mL)	2.00 ± 0.17 ^b^	11.02 ± 0.63 ^a^	8.80 ± 0.80 ^a^
Adiponectin (µg/mL)	12.34 ± 2.1 ^b^	21.14 ± 4.4 ^a^	17.52 ± 2.5 ^a^

Values are means ± SE. Different letters in a row indicate significant difference (*p* < 0.05) a > b > c. NEFA—non-esterified fatty acids.

## Data Availability

Data is available upon request.

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
