# Peer review of "The Role of the Unfolded Protein Response on Renal Lipogenesis in C57BL/6 Mice"

_biomolecules, 2021, doi:10.3390/biom11010073_

Round 1
Reviewer 1 Report
In this manuscript, Figueroa-Juarez and colleagues evaluated the impact of ER stress stimulation in kidney lipogenesis, by using C57BL/6 mice a model. Overall, this is an interesting and well-written manuscript. Experiments are well designed and results are clear, consistent, and well discussed. The final discussion of results is extensive and well contextualized.
However, results shown in figure 2 must include a control (vehicle-treated mice)
Reviewer 2 Report
I recommend publication of this article. The authors make interesting observations regarding induction of ER stress and pathological alterations in the kidney of mice treated with tunicamycin. The only correction I would like to be made is to add untreated control panels (C) to Fig 2 (Only 8 h and 24h treated panels are presented).
Round 2
Reviewer 1 Report
The authors provided the missing control and generated a new complete figure. Therefore, I now recommend the publication of the manuscript in Biomolecules